# COVID-19 Patients Presenting with Post-Intubation Upper Airway Complications: A Parallel Epidemic?

**DOI:** 10.3390/jcm11061719

**Published:** 2022-03-20

**Authors:** Grigoris Stratakos, Nektarios Anagnostopoulos, Rajaa Alsaggaf, Evangelia Koukaki, Katerina Bakiri, Philip Emmanouil, Charalampos Zisis, Konstantinos Vachlas, Christina Vourlakou, Antonia Koutsoukou

**Affiliations:** 1Interventional Pulmonology Unit and ICU of the 1st Respiratory Medicine Department National and Kapodistrian University of Athens, “Sotiria” Hospital, 115 27 Athens, Greece; aris.anag@yahoo.gr (N.A.); rz.alsaggaf@gmail.com (R.A.); e.koukaki@yahoo.gr (E.K.); aik.bakiri@gmail.com (K.B.); koutsoukou@yahoo.gr (A.K.); 2Interventional Pulmonology Unit, “Mediterraneo” Hospital, 166 75 Athens, Greece; philipemmanouil@hotmail.com; 3Thoracic Surgery Department, “Evangelismos” Hospital, 106 76 Athens, Greece; xzhs84@otenet.gr; 4Thoracic Surgery Department, “Sotiria” Hospital, 115 27 Athens, Greece; k.vahlas@gmail.com; 5Pathology Department, “Evangelismos” Hospital, 106 76 Athens, Greece; ch.vourlakou@yahoo.gr

**Keywords:** COVID-19, intubation, tracheostomy, airway complications, tracheal stenosis, tracheoesophageal fistula

## Abstract

During the current pandemic, we witnessed a rise of post-intubation tracheal stenosis (PITS) in patients intubated due to COVID-19. We prospectively analyzed data from patients referred to our institution during the last 18 months for severe symptomatic post-intubation upper airway complications. Interdisciplinary bronchoscopic and/or surgical management was offered. Twenty-three patients with PITS and/or tracheoesophageal fistulae were included. They had undergone 31.85 (±22.7) days of ICU hospitalization and 17.35 (±7.4) days of intubation. Tracheal stenoses were mostly complex, located in the subglottic or mid-tracheal area. A total of 83% of patients had fracture and distortion of the tracheal wall. Fifteen patients were initially treated with rigid bronchoscopic modalities and/or stent placement and eight patients with tracheal resection-anastomosis. Post-treatment relapse in two of the bronchoscopically treated patients required surgery, while two of the surgically treated patients required rigid bronchoscopy and stent placement. Transient, non-life-threatening post-treatment complications developed in 60% of patients and were all managed successfully. The histopathology of the resected tracheal specimens didn’t reveal specific alterations in comparison to pre-COVID-era PITS cases. Prolonged intubation, pronation maneuvers, oversized tubes or cuffs, and patient- or disease-specific factors may be pathogenically implicated. An increase of post-COVID PITS is anticipated. Careful prevention, early detection and effective management of these iatrogenic complications are warranted.

## 1. Introduction

The COVID-19 pandemic has dramatically changed the notions of severe respiratory insufficiency requiring intubation, mechanical ventilation (MV), and precautions against contamination from aerosol-generating procedures [1,2]. Upper airway complications after intubation or tracheostomy, although generally rare, still constitute an important factor of post-ICU morbidity and include damage to the vocal cords, laryngotracheal stenosis or granuloma formation, tracheomalacia, tracheoesophageal fistulae, and swallowing impairment [3,4,5]. Predisposing factors for the development of such complications, among others, include excessive cuff pressure and prolonged intubation before tracheostomy [6,7]. In the pre-COVID-19 era, performing a tracheostomy 7–14 days from endotracheal intubation and keeping the tracheal tube cuff pressure low (<30 cm H_2_O) has shown to decrease post-intubation complications [8,9].

In several laryngological reports [10,11] and in a recent position paper of the European Laryngological Society [12], it has been postulated that an unprecedented rise of iatrogenic laryngotracheal sequelae may develop in COVID-19 patients requiring intubation and MV. This is anticipated given the tremendous increase in the numbers of patients undergoing intubation and MV, and more so, since in these patients, the tracheostomy is often delayed for more than 2 weeks either due to concerns for high patient mortality, precaution against virus aerosolization, or the prolonged need for prone position MV [2,13]. It is indeed our recent experience that an unusually large number of patients, having recovered from a COVID-19 infection after ICU hospitalization and intubation, present with severe upper airway complications and have been referred for evaluation and management to our institution.

In this study, we aimed to prospectively describe the clinical and pathological characteristics of post-COVID/post-intubation patients in our cohort and report on their long-term management outcomes.

## 2. Methods

We prospectively analyzed data from all consecutive patients who were referred to our IP Unit for symptomatic upper airway complications following ICU admission and intubation due to a COVID-19 infection during the last 18 months. A negative COVID-19 PCR test and CT scan of the thorax were required before admission. After completing a relevant informed consent, patients’ clinical and anatomical characteristics of airway sequelae, as well as management and procedural related data and outcomes, were recorded. A diagnostic bronchoscopy was performed with a standard flexible video-bronchoscope (Olympus or Fujinon), during which the airway lesion was evaluated and measured. After a multidisciplinary discussion, and taking into consideration the patient’s preference, a management decision was taken either to refer the patient for tracheal surgery (tracheal resection and end-to-end anastomosis) or to treat the patient by rigid bronchoscopy, performing dilatation and/or—if needed—silicone stent placement.

In those patients who were admitted in emergency for severe dyspnea, a therapeutic intervention was planned and scheduled either on the same or on the following day. A rigid bronchoscopy was performed in the operating room under IV general anesthesia and muscle relaxation with manual jet ventilation. A mucosal web resection, dilatation, removal of granulation tissue, applying a combination of all available modalities (mechanical dilation, electrocautery, cryoablation), and/or a silicone stent placement were performed according to each patient’s specific indications and existing algorithms [14,15,16].

Furthermore, specimens of the tracheal wall segments resected from the patients who underwent tracheal surgery were examined for histological changes and were compared with other tracheal specimens resected from patients with post-intubation tracheal stenosis (PITS) of the pre-COVID-19 era to examine relevant similarities or differences.

The study protocol was submitted and was approved by the ethical scientific committee of our institution (1517/18-01-22)

### Statistical Analyses

An MS Excel database sheet was used for data collection and analysis, while statistical evaluation was performed with SPSS 28 (IBM, Armonk, NY, USA). Standard descriptive statistics are reported, including frequencies and percentages of the variables of interest.

## 3. Results

Twenty-three patients (15 men) were included in our study from June 2020 to January 2022. The demographic data of patients appear in Table 1.

Patients reported dyspnea due to PITS, 50 days ± 38 (mean ± SD) after their discharge from ICU. By the time of their readmission, a thorax CT was remarkable for unresolved lung infiltrates and ground glass opacities in only 6/23 patients (26%), while only 1 patient presented with hypoxemia and oxygen supplementation. Patients had several comorbidities, with the most prevalent being obesity (34.7%). The length of ICU hospitalization was 31.85 days ± 22.7 (mean ± SD) ranging from 6 to 98 days, whereas the duration of intubation was 17.35 days ± 7.4 (mean ± SD), ranging from 6 to 34 days with a median value of 15 days. Eleven patients (47.8%) underwent tracheostomy during their ICU stay. Nine of the tracheostomies were performed transcutaneously, while two were performed surgically. In three patients, the tracheostomy was still in place during admission in our unit as prior efforts to remove it had failed because of a relapse of dyspnea and stridor. In the majority of the cases, major hospital-associated pathogens i.e., *Pseudomonas* sp., *Klebsiella* pn., *Staphylococcus Aureus* and *Acinetobacter* sp. had been isolated from bronchial aspirates of the patients during their ICU hospitalization. Clinical and bronchoscopy findings are shown in Table 2.

Most of the patients presented with severe exertional dyspnea (mMRC > 3) and inspiratory stridor in 10 to 120 days (mean 32.8 ± 27 st. dev) after discharge from the ICU. Their thorax CT was remarkable for bilateral ground glass opacities and mosaic appearance of the lungs in as much as 56% of the patients. Tracheal cartilage fracture and distortion of the tracheal wall was evident in almost 83% of the patients. Figure 1.

Tracheal stenosis was severe in all patients and was located either in the subglottic area or in the mid trachea. In the vast majority of the patients, the stenosis was characterized as complex or mixed, as opposed to simple/web stenosis, in the sense that it involved the whole tracheal wall with alterations or fractures of the cartilages, distortion of the axis of the trachea, and focal malacia, while extending to an average length of 2.85 cm. In two patients, a tracheoesophageal fistula (TEF) had developed during intubation in the ICU and was revealed during a bronchoscopy (Figure 2).

Eight patients were referred to surgery, including one of the patients with TEF. A tracheal resection and an end-to-end anastomosis were performed following esophageal suturing for the TEF case. Post-surgery evaluation confirmed successful results in six of the patients, while in one patient, partial restenosis was evident, however not producing severe dyspnea, and in one patient, complete restenosis developed requiring rigid bronchoscopy dilatation and eventually stent placement.

The majority of the patients (15/23 or 65.2%), including the second case of TEF, were initially treated with rigid bronchoscopy dilatation and electrocautery or laser ablation of cicatricial tissue. Two more patients who had initially been treated surgically were later managed with rigid bronchoscopy modalities following post-surgery relapse. Stent placement was required overall in 12 patients, including the second patient with TEF, who underwent double stenting (tracheal and esophageal) since he was not amenable to surgery. In most cases, the hourglass-shaped silicone stent was implanted. Two of the patients who bared a tracheostomy cannula were decannulated and the tracheostomy was surgically closed following stent insertion, while in the third patient, tracheostomy was preserved due to its very high position right beneath the vocal cords and a Montgomery T-tube was placed to re-establish airway patency and vocation.

Post-treatment complications developed in nine of the patients treated by bronchoscopic means (60%): One patient developed a pneumothorax during the rigid bronchoscopy procedure directly after initiation of jet ventilation and immediately had a chest drain inserted. After a successful evacuation and resolution of the pneumothorax, a bronchoscopy was repeated the following week and a stent implantation took place successfully without further complications. The rest of the complicated patients had a relapse of tracheal stenosis on various occasions within one month after deployment of the stents due to pseudomembranous or thick purulent secretions obstructing the stent. These were eventually suctioned, and the thick or pseudomembranous material was removed using a cryoprobe. One patient presented with mucus retention and pseudomembranous formation causing tracheal stenosis and stent migration 4 weeks after the initial placement of the stent. Pseudomembranes were removed and the stent was repositioned successfully (Figure 3).

During follow-up, we had one death referring to the non-operable patient with TEF and double stenting who died of septic complications after 3 months of hospitalization. In two cases, the stents had to be removed due to continuous mucous accumulation and intense coughing, and these patients were subsequently referred to surgery achieving successful management. No other major complication was noted. The overall management histogram is shown in Figure 4.

An histology examination of the tracheal wall segments resected from the patients who underwent tracheal surgery revealed histological changes with fibrosis, inflammation, the healing process of the laryngotracheal mucosa, and degeneration and ischemic necrosis of the cartilaginous tracheal rings. In comparison with other tracheal specimens resected from patients with PITS of the pre-COVID-19 era, no specific differences were observed. Inflammatory infiltration consisted of a mixed population of chronic inflammatory cells involving a sufficient number of plasma cells. IgG/IgG4-related disease diagnostic findings were not established. Figure 5.

## 4. Discussion

Nowadays, post-intubation or post-tracheostomy tracheal injury is regarded as an uncommon iatrogenic complication and its incidence in some studies has been estimated to be as low as 4.9 cases per million of population per year [17]. Other studies have reported this complication in extremely variable estimates ranging from 1 in 1000 to 13% of the intubated ICU patients [4,7]. Acute laryngeal injury may occur much more often in patients with diabetes and larger body habitus or those who receive prolonged mechanical ventilation and an endotracheal tube greater than size 7.0 [18]. Nevertheless, it seems that the incidence of iatrogenic tracheal stenoses was much higher in the past and that the use of endotracheal tubes with low-pressure/high-volume cuffs did reduce the occurrence of this complication, although this still largely depends on the quality of the provided medical and nursing care [9,19,20].

Our facility is a large referral center for Interventional Pulmonology and central airway strictures management, receiving every year six to seven cases of post-intubation tracheal stenosis from different ICUs around the country. During the 5-year-period 2015–2019, we admitted and described 35 such patients [21], while since the COVID-19 pandemic outbreak, we have admitted an unprecedented high number of patients (23 in 18 months), who soon after discharge from the ICU developed stridor and exertional dyspnea due to post-intubation/post-tracheostomy airway stenosis and/or tracheoesophageal fistulae. The true incidence of PITS might be even higher as a number of post-COVID patients may not be referred for dyspnea/tracheal stenosis due to more debilitating COVID-related post-ICU sequelae. Other authors have also observed a similar increase in the incidence of these complications, which could either be attributed to the dramatic increase of intubated patients due to the pandemic or to other factors related to specific patients’ characteristics, to the disease itself, or ultimately to iatrogenic factors related with the hospitalization and the disease management under the current circumstances [22,23,24,25,26,27,28,29,30,31,32,33].

Early clinical reports have pointed out that COVID-19 patients experience extended stays in the ICU with prolonged orotracheal intubations often surpassing two weeks before discharge or tracheostomy being performed thus exposing them to increased risk for tracheal injury [12,22,28]. This has been attributed either to precautions against virus aerosolization or to the prolonged need for prone position MV. There have been reported several techniques using bronchoscopic and echographic guidance to facilitate and render safer the percutaneous tracheostomy procedure although these techniques are not universally applied [34,35].

Obesity being a risk factor for a dismal outcome in COVID-19 patients has also been associated with increased rates of developing post-intubation tracheal stenosis [23]. Indeed, in our cohort, patients had a slightly increased median time of orotracheal intubation of 15 days, which, however, in several cases even exceeded one month. Moreover, almost one-third of our patients were obese and mean ± st. dev. BMI was 29.4 ± 7.

In a recent laryngological case series [24], several COVID-19-related upper airway complications, including vocal fold immobility, granulation tissue formation, posterior glottic and subglottic stenosis, and posterior glottic diastasis have been associated with prolonged intubation in prone position and larger (>7.5) caliber orotracheal tubes. High incidence of post-extubation stridor and laryngeal injury among patients who underwent endotracheal intubation and tracheostomy during the COVID-19 pandemic has also been reported in other case series [25,26]. In a number of case reports, it was highlighted that common post-COVID-19 respiratory symptoms, such as cough and dyspnea after discharge from the hospital, were initially misdiagnosed and later attributed to post-intubation tracheal stenosis [27]. Prolonged intubation and prone position MV combined with obesity and diabetes have again been recognized as predisposing factors for the development of post-intubation tracheal stenosis in several small case series [28,29,30,31]. However, Fiacchini G. et al. [32] compared patients with and without COVID-19, all of whom had undergone prolonged (>14 days) MV. Almost half of the COVID-19 patients (47%) developed full thickness tracheal lesions or tracheoesophageal fistulas compared with only 2.2% of the non-COVID-19 patients, thus questioning the causative role of sole prolonged intubation and MV in the pathogenesis of these complications, pointing to more COVID-19- or patient-specific parameters.

With the exception of obesity and diabetes, comorbidities such as COPD, cardiovascular or other diseases were not recognized as risk factors for the development of this complication either in our study or in the literature. In our study, however, multi-resistant ICU pathogens were isolated in as much as 78% of the patients who developed tracheal injury, implying a possible role for microbial inflammation in the pathogenesis of these lesions.

Other mechanisms specifically related to COVID-19 pathophysiology and treatment may as well contribute to the observed high incidence of tracheal complications in this cohort of patients. Pronation maneuvers, which may increase the cuff pressure on the tracheal walls due to bending of the tube. could potentially lead to tracheal wall ischemic injury. Microvascular injury, ischemia, and necrosis of the laryngotracheal and esophageal mucosa, attributed to the prothrombotic state of COVID patients, could also produce necrosis and degeneration of the cartilaginous tracheal wall. Similar effects may be produced by the prolonged high doses of systemic corticosteroids, which are often used in these patients, hence modifying the healing process of the tracheal mucosa.

High viral replication in the tracheal epithelium could as well weaken the mucosa. Interestingly, in a recent autopsy study of tracheal epithelial cells, viral particles were found in them [33], while Ershadi R. et al. reported on a case of COVID-19-related tracheal stenosis due viral tracheitis in a patient who had never been intubated [36].

Apart from the impressive high incidence of tracheal injury, the most striking observation of our study was undoubtedly the high proportion of complex and subglottic stenosis: full thickness tracheal wall damage with fractured cartilages, distortion of airways axis, and intense inflammation in the area directly under the vocal cords were almost invariably observed. According to the classification of Freitag L. et al. [37], all of the stenosis were structural (usually of the scabbard type), located in the upper trachea, interestingly proximally to the area of the tube’s cuff.

All the above, even if histopathology did not reveal specific alterations, could potentially be attributed to a type of viral tracheitis. This, in the presence of other confounding factors either related to patients’ or management’s characteristics, could produce the observed airway damage [20]. In a recent histopathology report, a high localized density of IgG4 immunoglobulins-secreting plasma cells were found inside the fibrotic tissue of resected tracheal rings after post-COVID-19 PITS [38]. It could be speculated that the Th2 response induced by serious SARS-CoV-2 infections triggers in some patients a localized IgG4 hyperproduction with subsequent scarring sequelae, in particular at the level of the respiratory tract, even in the absence of IgG4-related disease.

In our study with the limited number of observations, although immunohistochemical stains showed abundant IgG+ plasma cells, some of which were also IgG4+, the findings were similar between post-COVID and non-COVID PITS. An IgG4-RD or any other specific COVID-related pathology could not be confirmed.

Nevertheless, irrespective of the underlying tracheal pathological alterations and local inflammation process, the unprecedented intubation protocol used on airway management of COVID-19 patients, performed by emotionally and physically exhausted caring physicians wearing cumbersome personal protective equipment and face shields, could potentially increase the possibility of (unconscious and often unreported) immediate tracheal injury. To make things worse, most of the COVID patients presenting with respiratory insufficiency are intubated late in the course of their disease after all other non-invasive measures have been exhausted, thus resulting in emergency intubation conditions.

The management of PITS (regardless of the disease for which intubation and MV was required) is still a matter of debate, and there is no clear consensus about the best treatment strategy. Gallucio et al. confirmed that interventional bronchoscopic procedures are a valid therapeutic option in selected cases with PITS, in both simple and complex stenosis following a validated algorithm [14]. They reported a success rate of 96% in simple stenosis and 69% in complex stenosis. Our group has published analogous results following a similar clinical algorithm [15,16].

In COVID-19-related cases, some authors have suggested managing PITS endoscopically with balloon dilation, intralesional corticosteroid injection, and consecutive endoscopic assessment [23,28]. Moreover, during the pandemic era, it has been recommended, that tracheal surgery should be avoided as the primary choice in COVID-19 patients and be reserved only for high-complex stenosis or any other stenosis having relapsed after extensive endoscopic treatment [28,39]. In the current study dealing with severely debilitated patients with important dyspnea and advanced neuromuscular deconditioning, initial surgical management was suggested in only 35% of these patients, two of which relapsed after surgery and had to undergo bronchoscopic treatment to regain airway patency. Post-bronchoscopic dilatation/stent placement complications (mainly mucus accumulation and pseudomembranes obstructing the stent) developed in 60% of patients requiring an average of 3.5 review flexible bronchoscopic sessions per patient within the first 3 months after initial treatment. In two of them, further definite surgical treatment had to be administered. These results underline the necessity for close interdisciplinary cooperation between interventional pulmonologists and thoracic surgeons as well as the importance of careful follow-up of these patients who have to be regularly examined with spirometry and flexible bronchoscopy for at least the first trimester following their treatment. Having said this, rationalizing the use of all the available interventional and surgical modalities, the overall long-term results were satisfactory in the vast majority of our patients.

## 5. Conclusions

During the current pandemic, we witness an unprecedented increase of post-intubation tracheal stenosis in COVID-19 patients discharged from ICU. It is highly probable that in the near future interventional pulmonologists and thoracic surgeons will be called to manage a rising number of such upper airway complications. A multi-disciplinary approach making use of rigid bronchoscopic modalities regaining airway patency with or without stent placement combined with surgical tracheal resection and anastomosis to those who are eligible for surgery is highly effective. Patients presenting with dyspnea and/or stridor after discharge from the ICU should be examined with spirometry and CT imaging for laryngotracheal stenosis. Extreme caution is warranted in the airway management of these patients while in the ICU to prevent such post intubation airway sequelae. Further studies may clarify whether they are due to specific COVID-19 and/or patients’ clinical and epidemiological features or if they are correlated to prolonged ICU intubation, delayed tracheostomy, oversized tubes and cuffs or eventually to iatrogenic trauma during emergency intubation.

## Figures and Tables

**Figure 1 jcm-11-01719-f001:**
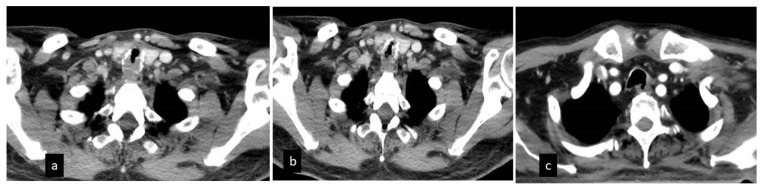
CT of the thorax depicting upper trachea stenosis with posterior wall thickening (**a**), anterior wall cartilage fracture (**b**) and tracheoesophageal fistula (**c**).

**Figure 2 jcm-11-01719-f002:**
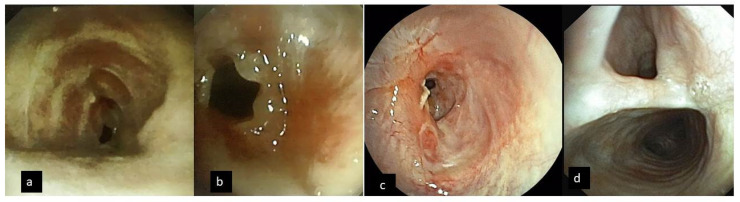
Bronchoscopic view of severe complex upper trachea stenosis (**a**–**c**) and tracheoesophageal fistula (**d**).

**Figure 3 jcm-11-01719-f003:**
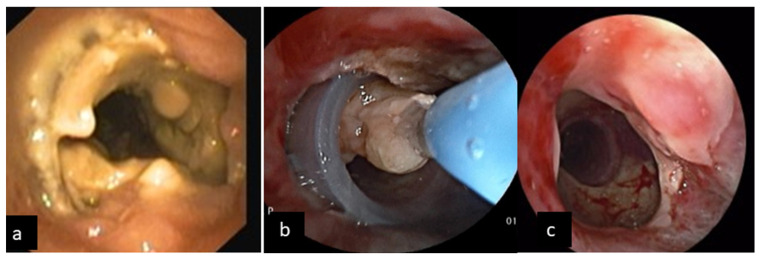
Bronchoscopic view of complicated tracheal stents with obstruction due to biofilm and thick purulent mucous accumulation (**a**), removal of sticky secretions making use of cryoprobe (**b**) and peripheral stent migration (**c**).

**Figure 4 jcm-11-01719-f004:**
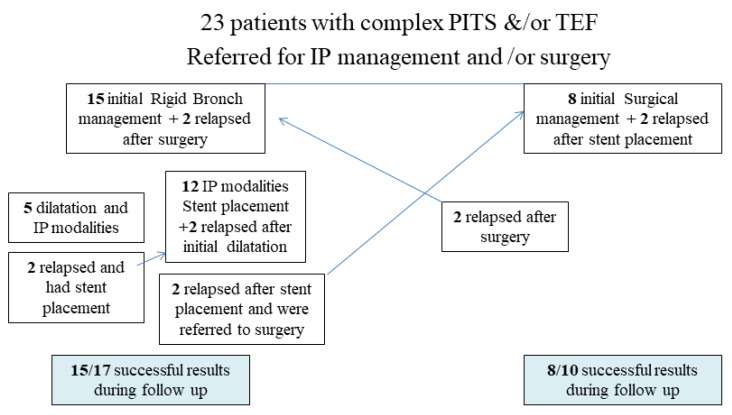
Histogram depicting the multi-disciplinary management of our patients and their long-term outcome.

**Figure 5 jcm-11-01719-f005:**
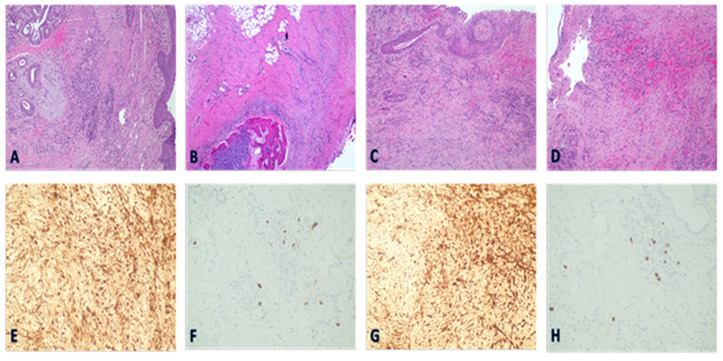
Histological section of the trachea with fibrosis and plasma cell infiltration (hematoxylin-eosin; ×20 magnification). Upper row: Representative histological micrographs comparing the lateral part of subglottic tracheal stenosis between a patient with post-COVID stenosis (**A**,**B**) and a patient with non-COVID stenosis (**C**,**D**) (hematoxylin-eosin staining; ×20 magnification). The findings are similar. (**A**). Squamous metaplasia, thickening, and scarring of submucosa, considerable interstitial fibrosis. (**B**). Submucosal vessels dilatation, hyperemia, and granulation tissue proliferate. (**C**,**D**). Degeneration and ischemic necrosis of the cartilaginous tracheal rings. Lower row: on immunohistochemistry, plasma cells are mainly IgG-secreting elements, with a low tissue density of IgG4 subclass (IHC ×20 objective). The findings are similar between post-COVID (**E**,**F**) and non-COVID patients (**G**,**H**). Immunohistochemical stains show abundant IgG+ plasma cells, some of which are also IgG4+. The IgG4+ plasma cells exhibit a patchy distribution (IgG4, immunoperoxidase, original magnification ×200). A definite histopathologic diagnosis of IgG4-RD is not proven. Abbreviation: IgG, IgG4, immunoglobulin G and immunoglobulin G4.

**Table 1 jcm-11-01719-t001:** Patients’ characteristics.

Demographics	N (%) or Mean ± St Dev
Age (years, mean ± st dev)	58 ± 9.5
Gender (M/F)	15/8
**Comorbidities**	
BMI (mean ± st dev)	29.4 ± 7
Obesity (BMI > 30) (%)	8/23 (34.7%)
Smoking History	5/23 (21.7%)
Cardiovascular (Hypertension, Coronary dis)	5/23 (21.7%)
Diabetes mellitus	4/23 (17.4%)
Sleep Apnea Hypopnea s.	3/23 (13%)
Other (COPD, Asthma, GERD etc)	None
**COVID related**	
Days of ICU hospitalization [mean ± st dev, (min–max), median]	31.85 ± 22.7, (6–98), 25.5
Days of orotracheal intubation before tracheostomy was performed. [mean ± st dev, (min–max), median]	17.35 ± 7.4, (6–34), 15
Tracheostomy, TC/surgical	11/23 (47.8%), 9/2
ICU pathogens (*Pseudomonas* sp., *Klebsiella pn.*, *Staph.* *Aureus*, *Acinetobacter* sp.) isolated from aspirates.	18/23 (78.3%)

**Table 2 jcm-11-01719-t002:** Airway sequelae and bronchoscopy findings.

Clinical/Radiological	N (%) or Mean ± St Dev
Dyspnea mMRC score	3.04 ± 0.97
Inspiratory stridor	20/23 (86.7%)
Evidence tracheal cartilage fracture in the CT	19/23 (82.6%)
Evidence of residual bilateral ground glass opacities	13/23 (56.5%)
**Bronchoscopy findings**	
Tracheal stenosis %	84.45% ± 8.3%
Length of stenosis (cm)	2.85 ± 0.9
Subglottic/mid trachea	12/11
Complex or mixed/simple web	21/2
Distortion of the airway due to anterior wall cartilage fracture	19/23 (82.6%)
Excessive Dynamic Airway Collapse	5/23 (21.7%)
Tracheo-esophageal fistula	2/23 (8.7%)
Foreign body aspiration	1/23 (4.3%)

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
