# Peer review of "COVID-19 Patients Presenting with Post-Intubation Upper Airway Complications: A Parallel Epidemic?"

_jcm, 2022, doi:10.3390/jcm11061719_

Round 1

Reviewer 1 Report

The article is interesting, well prepared and regards a very important issue of a postintubation tracheal stenosis after COVID

Author Response

The article is interesting, well prepared and regards a very important issue of a postintubation tracheal stenosis after COVID.

Thank you very much for your kind consideration

Reviewer 2 Report

This paper is a hot topic: I have some suggestions

methods: please explain better the technique of tracheostomy? Is there an expert team to perform the procedure? did you use the echographic check before? Why did you not use it? It could help you 

How many bronchoscopic did you perform during the length of stay?

  • DOI: 10.1016/j.athoracsur.2020.06.011

You provided histological findings: very interesting. Do you measure fewer complications in patients treated with interleukin inhibitors?  

Author Response

This paper is a hot topic: I have some suggestions.methods: please explain better the technique of tracheostomy? Is there an expert team to perform the procedure? did you use the echographic check before? Why did you not use it? It could help you. How many bronchoscopic did you perform during the length of stay?

Dear reviewer thank you very much for your kind consideration. As most of the cases with post intubation stenosis were referred to our interventional unit from many other hospitals all around Greece and that was several months after their discharge from the ICU, we do not have enough data on the number of the bronchoscopies performed or the exact methodology of tracheostomy performance. Echographic check is not usually applied in these cases. We have added though a comment  with your suggested references in the Discussion section. (page 7: " There have been reported several techniques using bronchoscopic and echographic guidance to facilitate and render safer the percutaneous tracheostomy procedure although these techniques are not universally applied. [37-38]"

  • DOI: 10.1016/j.athoracsur.2020.06.011

This reference has been added as described above.

You provided histological findings: very interesting. Do you measure fewer complications in patients treated with interleukin inhibitors?  

Thank you again for your kind consideration. Unfortunately we have no data at all to compare patients who received or did not receive IL inhibitors during their stay in the ICU. This is a real life study focusing on patients presenting with this complication  and was not designed to compare medical regimens administered.

Reviewer 3 Report

Dear Author(s),
This study on upper airway complications in COVID-19 patients followed in the intensive care unit is a remarkable study. However, the authors could have developed this study with a larger number of patients under pandemic conditions. If the conditions for providing this were not met, they should state it under the heading "Limitation of Study" at the end of the study. The purpose, design, results and discussion of the results are clear and understandable.
In addition, it would be appropriate to write the microorganism names specified in the article in italics due to the spelling rules.

Best Regards

Author Response

This study on upper airway complications in COVID-19 patients followed in the intensive care unit is a remarkable study. However, the authors could have developed this study with a larger number of patients under pandemic conditions. If the conditions for providing this were not met, they should state it under the heading "Limitation of Study" at the end of the study. The purpose, design, results and discussion of the results are clear and understandable.

Dear Reviewer we are grateful for your kind consideration. As stated in the methodology section, we included all the patients presenting with post COVID/post-intubation stenosis during the last 18 months who were reeferred to our IP unit from many Hospitals in Greece. We do not have data on the current incidence of this complication which used to be very rare before COVID outbreak. To the best of our knowledge this is the largest cohort of these patients reported.
In addition, it would be appropriate to write the microorganism names specified in the article in italics due to the spelling rules.

Thank you for your suggestion this has been corrected.